# Genome, Metabolism, or Immunity: Which Is the Primary Decider of Pancreatic Cancer Fate through Non-Apoptotic Cell Death?

**DOI:** 10.3390/biomedicines11102792

**Published:** 2023-10-14

**Authors:** Erfaneh Barar, Jiaqi Shi

**Affiliations:** 1Liver and Pancreatobiliary Diseases Research Center, Digestive Disease Research Institute, Shariati Hospital, Tehran University of Medical Sciences, Tehran 1416753955, Iran; 2Department of Pathology & Clinical Labs, Rogel Cancer Center, Center for RNA Biomedicine, University of Michigan, Ann Arbor, MI 48109, USA

**Keywords:** pancreatic ductal adenocarcinoma, apoptosis, ferroptosis, necroptosis, pyroptosis

## Abstract

Pancreatic ductal adenocarcinoma (PDAC) is a solid tumor characterized by poor prognosis and resistance to treatment. Resistance to apoptosis, a cell death process, and anti-apoptotic mechanisms, are some of the hallmarks of cancer. Exploring non-apoptotic cell death mechanisms provides an opportunity to overcome apoptosis resistance in PDAC. Several recent studies evaluated ferroptosis, necroptosis, and pyroptosis as the non-apoptotic cell death processes in PDAC that play a crucial role in the prognosis and treatment of this disease. Ferroptosis, necroptosis, and pyroptosis play a crucial role in PDAC development via several signaling pathways, gene expression, and immunity regulation. This review summarizes the current understanding of how ferroptosis, necroptosis, and pyroptosis interact with signaling pathways, the genome, the immune system, the metabolism, and other factors in the prognosis and treatment of PDAC.

## 1. Introduction

Pancreatic cancer is a poor-prognosis cancer with an aggressive nature [1]. The number of cases of pancreatic ductal adenocarcinoma (PDAC), the majorly prevalent form of pancreatic cancer, is anticipated to increase, mainly due to the average population age increasing [1,2]. Despite decades of improvement, there is still a low 5-year survival rate for patients with PDAC, which has only increased to 12% [3]. Furthermore, PDAC is considered a challenging and poor-prognostic cancer to treat due to its late detection and resistance to chemotherapy [4].

Of symptomatic PDAC patients, 80% are beyond surgical possibilities because of distant metastasis or the involvement of vital structures [5,6,7,8,9]. As a result, chemotherapy plays a crucial role in treating PDAC [6,7]. During advanced stages of PDAC, gemcitabine (GEM) in combination with nanoalbumin-bound paclitaxel (nab-PTX) or FOLFRINOX (a combination of leucovorin, 5-FU, irinotecan, and oxaliplatin) is the cornerstone of chemotherapy based on patients’ conditions [8,10]. Apoptosis-related mechanisms can be modulated by different immunogenetic factors in PDAC, increasing chemoresistance [9]. Regarding apoptotic immunogenetic modulators, most PDAC patients harbor at least one of two frequently mutated genes, *KRAS* and *TP53* [11,12]. The research demonstrated a correlation between the co-occurrence of *KRAS*-*TP53* alterations and immune-excluded microenvironments, resistance to chemotherapy, and unfavorable survival outcomes in PDAC [13]. *KRAS* mutations can inhibit apoptosis by downregulating proteins that promote apoptosis, such as BAX, and upregulating proteins that inhibit apoptosis, such as BCL-2. *TP53* encodes a checkpoint protein, p53, which functions as a tumor suppressor and is transcriptionally essential for BAX [11,12]. The BCL-2 protein is also upregulated by activated pancreatic stellate cells (PSCs), which are crucial tumor microenvironment (TME) components. By preventing apoptosis, PSCs play a vital role in PDAC chemoresistance [14].

In addition to PSCs, the TME comprises other immune cells, cancer-associated fibroblasts (CAFs), and networks, forming a dense, multilayered structure [15,16,17]. The TME, as a double-edged sword in PDAC progression, contributes to PDAC tumor heterogeneity in patients [18]. The immune-dense barrier of the TME plays a pro-tumor role by impeding drug penetration into PDAC and recruiting immune cells, including M2-polarized macrophages, neutrophils, and exhausted tumor-infiltrating lymphocytes (TIL) [15,16,17]. In contrast, the TME demonstrates anti-tumor activity by engaging cytotoxic T cells or signaling pathways that interact with immune cells (including nuclear factor kappa-light-chain-enhancers of activated B cells (NF-κB)) [15,16,17]. However, NF-κB is known for promoting cancer cell survival by controlling the expression of genes involved in immune functions, such as cytokines and chemokines, and regulating anti-apoptotic genes (BCL-2 and BCL-xL) [15,19,20]. A further effect of NF-κB is that it promotes the migration of CAFs within the TME, increasing the risk of chemoresistance and subsequent cancer progression [17,21]. Notably, oncogenic KRAS signaling drives NF-κB activation in PDAC [20].

Additionally, PDAC exhibits hypoxia as a significant aspect of the TME. Through different mechanisms, hypoxia causes malignant PDAC features, affecting prognosis. These include triggering genes related to angiogenesis, glycolysis, and various molecules and signaling pathways that cause invasion and drug resistance in PDAC [22,23,24]. Based on these studies, Chen et al. designed a prognostic risk score model in PDAC, including seven hypoxia- and immune-associated signature genes (*S100A16*, *PPP3CA*, *SEMA3C*, *PLAU*, *IL18*, *GDF11*, and *NR0B1*), and classified patients according to their risk level. The high-risk population exhibited different immunocyte infiltration states and mutation spectra and lower immune scores, stromal scores, and immune checkpoint expression, such as anti-Programmed Death 1 ligand (anti-PD-L1). Conversely, immunotherapy may benefit patients with low-risk scores and high immune checkpoint expression [25].

An astonishing study by Wang et al. indicated that the immune system and metabolic pathway are closely intertwined, with implications for prognosis and therapeutic response. Through single-cell RNA sequencing, considerable heterogeneity was detected in CAFs, immune cells, and ductal cancer cells in two PDAC types, classified as the dense type (high desmoplasia) and loose type (low desmoplasia). The loose-type PDACs possessed a distinct subtype of CAFs, called meCAFs, distinguished by a markedly activated metabolic profile. A high level of glycolysis was observed in meCAFs, while oxidative phosphorylation was utilized instead of glycolysis in cancer cells. It is important to note that patients with overexpression of meCAFs were mainly prone to undergo metastasis and faced a poorer prognosis. However, there was a dramatic improvement in their response to immunotherapy [26].

It was concluded from these studies that the conversation among genetics, immunity, metabolism, and signaling pathways contributes to enhanced chemoresistance and adverse prognosis in PDAC, in part through apoptosis-resistance mechanisms. In this regard, improving our understanding of non-apoptotic cell death processes enables us to overcome chemoresistance. Therefore, it is imperative to understand the dynamic interaction among genetics, immunity, metabolism, and signaling pathways in significant types of non-apoptotic cell death in PDAC, such as ferroptosis, necroptosis, and pyroptosis [27]. This review summarizes how ferroptosis, necroptosis, and pyroptosis interact with critical factors, including the genome, the immune system, the metabolic system, and signaling pathways, for the prognosis and treatment of PDAC (Table 1).

## 2. Ferroptosis

The overperoxidation of lipids triggers ferroptosis, a non-apoptotic immune cell death [34]. Both oxidative and antioxidant mechanisms contribute to lipid peroxidation, ultimately leading to iron toxicity [35]. Iron-dependent cell death, ferroptosis, was found to play a significant role in iron-rich tumors such as PDAC, based on the tumor stage and TME [35,36]. Releasing damage-associated molecular patterns (DAMPs) induced by ferroptosis activates immune responses within the TME. Although the DAMPs released by ferroptosis were found to promote tumor cell growth through macrophage polarization in the TME of PDAC [36], the absence of ferroptosis was found to contribute to PDAC tumorigenesis in mouse models [34]. Aside from the dual role ferroptosis plays in PDAC, the potential role of drug-induced ferroptosis in controlling PDACs has gained increasing attention, making ferroptosis a potentially promising therapeutic strategy [34,36]. Therefore, addressing ferroptosis-related signaling pathways, the genome, the metabolism, and immune systems can open new avenues for improving PDAC prognosis.

### 2.1. Signaling Pathway

Ferroptosis is mainly caused by iron-dependent lipid peroxidation, which depends on glutathione (GSH) synthesis, reactive oxidative species (ROS) production, lipid peroxidation, and iron accumulation [35]. Several components are involved in the extrinsic pathway, including System XC^−^, GSH, glutathione peroxidase 4 (GPX4), and glutathione disulfide (GSSG), with GSH, ultimately, determining the outcome of ferroptosis. Antiporter protein System XC^−^ replaces extracellular oxidized cysteine (cystine) with intracellular glutamate. As a result of the systemic XC^−^ antiporter’s ability to induce ROS production through GSH synthesis, it promotes ferroptosis. Conversely, the conversion of phospholipid hydroperoxides (PLOOH) into corresponding phospholipid alcohols and GSH into GSSG by GPX4 (the antioxidant enzyme) prevents the production of ROS and lipid peroxidation, contributing to ferroptosis inhibition [37,38,39,40,41]. Thus, interruption of GPX4 and subsequent suppression of the system XC^−^-GSH-GPX4 axis results in ferroptosis [42]. As one of the primary arms of ferroptosis, iron is crucial for recruiting diverse agents to promote intrinsic pathways. Iron accumulation produces ROS through the Fenton reaction [43]. Ferroptosis may also result from the destruction of ferritin, a protein that intracellularly stores iron, and the subsequent release of this iron by autophagy. It was demonstrated that autophagy receptors, such as nuclear receptor coactivator 4 (NCOA4) and sequestosome 1 (SQSTM1), facilitate the degradation of ferritin or SLC40A1 (an iron transporter) in PDAC cells [44,45,46]. Although PDAC depends on autophagy as a survival mechanism, triggering ferroptosis mediated by autophagy (ferritinophagy) likely kills established PDAC cells [47].

Heat shock proteins (HSP) are molecular chaperones with a critical function in PDAC tumor growth by orchestrating endoplasmic reticulum stress, protein modulation, and ferroptosis [34]. The endoplasmic reticulum (ER) is a vital organelle in ferroptosis, whose homeostasis is controlled by Heat Shock Protein Family A (HSP70) Member 5 (HSPA5). Endoplasmic reticulum stress-related transcription factor 4 (ATF4) mediates the expression of HSPA5, a negative ferroptosis regulator, by enhancing GPX4 stability. Consequently, suppressing the ATF4-HSPA5-GPX4 pathway enhances ferroptosis in PDAC cells [48,49]. Furthermore, by producing enzymes such as Acyl-CoA synthetase long-chain family member 4 (ACSL4), ER enables the augmentation of lipid peroxidation and enhances ferroptosis susceptibility in PDAC [50]. As another organelle that modulates lipid peroxidation and produces ROS, mitochondria can also have ferroptotic effects [51,52,53,54]. In PDAC cells, pyruvate oxidation promotes ferroptosis by activating acetyl- CoA carboxylase alpha (ACACA)- and fatty acid synthase (FASN)-mediated fatty acid synthesis and subsequent ALOX5-dependent lipid peroxidation [51] (Figure 1).

### 2.2. Immunogenetics

#### 2.2.1. Ferroptosis-Related Genes (FRGs) as Risk Models in PDAC Prognosis

The genome, acting as a cell leader, modulates several different pathways (for example, protein expression, signaling pathways, and the immune system), significantly impacting PDAC metastasis, prognosis, and resistance to treatment [55]. Thus, it is imperative to translate the increased knowledge of tumor genetics and genomics into clinically useful gene signatures. Various ferroptosis-related genes (FRGs) are identified through the Cancer Genome Atlas (TCGA) and different genome analyses, which are utilized in developing risk models for pancreatic cancer (Table 2). Feng et al. developed a risk model based on five FRGs, where *CAV1*, *DDIT4*, *SRXN1*, *and TFAP2C* indicate better survival, whereas *SLC40A1* indicates a more adverse outcome [56]. Using *ZNF419*, *TUBE1*, *STEAP3*, *SLC1A4*, *RRM2*, *PTGS2*, *MT1G*, *MAP3K5*, *DDIT4*, *CAPG*, *CAV1*, *BAP1*, *AURKA*, and *ATG4D*, Jiang et al. designed a prognostic FRGs risk model, in which it was shown that expression levels of *PTGS2*, *RRM2*, *AURKA*, *CAV1*, *MAP3K5*, and *STEAPS* are higher in tumor samples, and the upregulation of *PTGS2* and the downregulation of *MT1G, TUBE1*, and *ATG4D* might contribute to tumorigenesis and poor outcomes [57]. In another study, a gene signature was demonstrated to have powerful predictive capabilities for overall and disease-free survival in PDAC. The genes *ASPH*, *DDX10*, *NR0B2*, *BLOC1S3*, *FAM83A*, *SLAMF6*, and *PPM1H* do not overlap with other prognostic gene signatures for PDAC. They could, therefore, complement the existing staging system for prognosis evaluation and treatment planning [58]. An independent predictive model was constructed based on four FRGs (*ENPP2*, *ATG4D*, *SLC2A1*, and *MAP3K5*), in which the high-risk group responded better to chemotherapy than the low-risk group. Notably, this prognostic model may affect immune cells and checkpoints, demonstrating the connection between FRGs and the immune system [59]. According to Chen et al., there was a possible correlation between FRGs, signaling pathways, and metabolic pathways. Their study demonstrated that ferroptosis-related long noncoding RNAs are prognostic biomarkers for patients with PDAC by utilizing *SLC16A1-AS1*, *SETBP1-DT*, *ZNF93-AS1*, *SLC25A5-AS1*, *AC073896.2*, *LINC00242*, *PXN-AS1*, and *AC036176*. It was found that the low-risk subgroup is enriched in ferroptosis-related pathways (fatty acid metabolism and oxidative phosphorylation), the survival cancer pathway (PI3K-AKT-mTOR signaling), and organelles (peroxisome and lysosome) [60]. Considering these studies, there appears to be a connection among genomes, the metabolism, signaling pathways, and immune systems, which is further explored in the following sections.

#### 2.2.2. Interactions among Genomes, Immune System, Metabolism, and Signaling Pathways in PDAC

PDAC is characterized by the interplay among genomes, inflammation, and metabolic reprogramming, which are linked to cancer progression, tumor stages, and response to treatment [35,36,80]. For example, Shang et al. demonstrated that TRIM11, which regulates immune-related signaling pathways, significantly contributes to ferritinophagy and gemcitabine resistance in PDAC [81]. A significant role of the genome concerning signaling pathways, metabolic pathways, and the immune system is revealed by the differences in these pathways within *KRAS*- or *TP53*-mutated PDAC cells. *KRAS*-mutant cancer cells inhibit ferroptosis by regulating metabolic pathways, which includes modulating the metabolism of fatty acids, amino acids, or glucose [82,83,84,85]. In PDAC cell lines harboring either *KRAS* and *TP53* double mutations or *TP53* single mutations, MMRi62, a novel ferroptosis-inducer molecule, inhibits tumor growth and migration. As a result of modulating ferritinophagy-related factors (the downregulation of NCOA4 and the degradation of FTH1) and mutant *TP53*, MMRi62 stimulates ferroptosis and anticancer effects [86]. In *KRAS*-mutant PDAC, inflammatory mediators (e.g., cytokines, DAMPs, and immune cells) are involved in pro- or anti-tumor gene modulation to sustain a favorable inflammatory TME for tumor growth and development [87]. Autophagy influences iron metabolism and immunity, contributing to the progression of PDAC [88,89]. According to Mukhopadhyay and colleagues, the administration of a low-iron diet in vivo enhanced the response to autophagy inhibition therapy in PDAC [89]. Iron homeostasis is sustained by autophagy in PDAC. Therefore, autophagy suppression by reducing labile iron concentrations leads to modifications in mitochondrial metabolism in PDAC [89]. Furthermore, CAFs in the TME provide bioavailable iron to PDAC cells, which promotes resistance to autophagy suppression. A high level of autophagy in PDAC helps them survive in the dense TME [89]. Furthermore, ferritinophagy mediated by NCOA4 is also upregulated in PDAC [88].

DNAJB11 is a co-chaperone for HSPA5 with a dual effect on PDAC progression [90]. DNAJB11 could regulate epidermal growth factor receptor (EGFR) expression and initiate the subsequent mitogen-activated protein kinase (MAPK) signaling pathway, ultimately promoting cancer. However, DNAJB11 regulates ER stress and negatively controls the unfolded protein response (UPR) signaling pathway [90]. As a result of the UPR regulation of cytokines and PSCs, an optimal TME could be shaped for PDAC progression [91]. Although HSP90 generally induces ferroptosis through the degradation of GPX4 [92], it enhances the pro-tumor resistance of the TME in PDAC by directly improving the growth of PSCs and CAFs in vitro [93]. By inhibiting HSP90, it may be possible to reduce inflammatory signals within the TME, thereby enhancing its sensitivity to immunotherapy [93]. Mitochondrial-mediated ROS production plays a crucial role in the tumorigenicity of KRAS-dependent tumors such as PDAC. Despite the dual dose-dependent role of ROS in pro- and anti-tumor progression, toxic levels suppress tumor growth by inducing cell death [52,53,54]. PDAC cells upregulate metabolic programs that prevent the reduction in intracellular ROS or detoxify lipid ROS to inhibit ferroptosis [82,94]. Aspartate aminotransferase (GOT1) plays an essential role in PDAC as a vital member of the metabolic pathway. Aside from maintaining ROS levels to promote *KRAS*-mutant PDAC, GOT1 inhibits ferroptosis by detoxifying levels of ROS in PDAC [82,83]. Eliminating cystine import, GSH synthesis, or GPX4 in synergy with GOT1 could cause ferroptosis. GOT1 inhibition impairs the ferroptosis–inducer–mitochondrial metabolism in the PDAC cell line [95]. Lipid ROS are cleaned by one of the XC- system members, solute carrier family 7 member 11 (SLC7A11), with cysteine uptake. In CAFs, SLC7A11 is upregulated, promoting PDAC cell growth by suppressing ferroptosis [96]. In addition, the correlation between the signaling pathway and gene expression is predominant by showing that tumorigenesis in *KRAS*-mutant PDAC in vitro can be inhibited by deleting ferroptosis-related signaling pathway factors such as SLC7A1 or GPX4 [36].

### 2.3. Treatment

#### 2.3.1. Drugs That Modulate Signaling Pathways through Ferroptosis

Ferroptosis-related treatments include drugs that alter ferroptosis substances, signaling pathways, the immune system, and the function of organelles (Table 3). The system XC-GSH-GPX4 axis can be affected by different experimental drugs. Cysteinase is an XC^-^ system inhibitor that depletes cysteine and cystine [40]. Small molecules that activate ferroptosis through intrinsic pathways are important signaling pathway modulators. Imidazole Ketone Erastin (IKE), a member of the ferroptosis intrinsic pathway, is another XC^-^ system inhibitor [40]. In addition to RSL3, another ferroptosis activator that promotes autophagy, Sirolimus (also known as rapamycin), induces ferroptosis through the autophagy-mediated degradation of GPX4 in vivo and in vitro [97]. Naturally derived ferroptosis-inducer drugs increasing intracellular ROS and/or iron accumulation in PDAC include Artesunate (ART) (ROS and intracellular iron), Piperlongumine (PL) (ROS), and Ruscogenin (iron) [98,99,100].

#### 2.3.2. Drugs That Modulate Organelle Functions through Ferroptosis

ER and mitochondria, as two vital organelles for ferroptosis cell death, are affected by different preclinical and clinical drugs in PDAC. In addition to the conventional triggers of ferroptosis (erastin and RSL3) resulting in autophagy, Zalcitabine (an HIV drug) stimulates autophagy-mediated ferroptosis (ferritinophagy) by affecting ER and mitochondria. The degradation of mitochondrial transcription factor A (TFAM) in response to Zalcitabine generates an increase in mitochondrial DNA stress and ROS levels. As a result of ROS accumulation, ferritin is degraded via autophagy, which, in turn, leads to the formation of lipid peroxides and ferroptosis. Mitochondrial stress is detected by the ER protein (stimulator of interferon genes (STING)) as well as elevated ROS, resulting in lipid peroxidation and, eventually, autophagy-mediated ferroptosis [104,105].

#### 2.3.3. Combination Therapy

Although inducing ferroptosis improved cancer treatment in animal models, many open questions remain. The deletion of *GPX4*, one of the primary anti-ferroptosis arms, in PDAC precursor lesion pancreatic intraepithelial neoplasia (PanIN) did not result in ferroptotic cell death, indicating that *KRAS*-mutated cells possess a protective arm that prevents ferroptosis [111,112]. According to the most recent study, ferroptosis suppressor protein 1 (FSP1) is an important protective factor upregulated in *KRAS*-mutant cells. Thus, combination therapy strategies were shown to improve outcomes with ferroptosis activator and FSP1 inhibitor in PDAC, indicating the importance of combination therapy [112].

GEM is a cornerstone of most combination therapies for PDAC, and other drugs are evaluated based on their ability to increase GEM sensitivity. As a result of GEM treatment, carbonyl reductase 1 (CBR1), an antioxidant enzyme, is upregulated. Since CBR1 levels are directly associated with PDAC chemoresistance and poor prognosis, the administration of chrysin (a CBR1 inhibitor) enhances GEM chemosensitivity in vitro and in vivo through the accumulation of ROS [106]. Other agents induce GEM sensitivity by modulating the SLC family, the HSP family, ferroptosis substances, the system XC^—^GSH-GPX4 axis, and organelle function. Lesinurad (an inhibitor of pan-SLC22A) reduces metastasis and GEM chemoresistance in mouse models of PDAC, thus identifying novel vulnerabilities in human PDAC [107]. The ability of epigallocatechin gallate (EGCG) and sulfasalazine (SSZ) to inhibit HSPA5 promotes ferroptosis and improves sensitivity to GEM chemotherapy by weakening the binding between HSPA5 and GPX4 as well as destabilizing GPX4 [48]. Additionally, incorporating SSZ into the combination of PL and cotylenin A (a growth regulator), which themselves induce ferroptosis in MiaPaCa-2 and PANC-1 PDAC cells by increasing ROS production, synergistically hinders these cell lines’ survival [100]. As a result of inhibiting SLC7A11, SSZ diminished the viability of the HPAF-II PDAC cell line treated with docosahexaenoic acid. By modulating the GSH level and restricting nucleotide synthesis, docosahexaenoic acid prevents *KRAS*/*TP53* double-mutant PDAC cells from multiplying and triggering apoptosis in HPAF-II cells. The combination of docosahexaenoic acid and GEM in this cell line effectively induced oxidative stress and cell death [108]. Nuclear protein 1 (NUPR1) is a well-known GEM resistance inducer, and its subsequent significance in the development of PDAC cannot be overlooked [113]. There is evidence that ZZW-115, one of the most prominent NUPR1 inhibitors, induces ferroptosis in PDAC through the modulation of organelle function (ER and mitochondria) and metabolic shifts to glycolysis. Mitochondrial-dependent ferroptosis mediated by ZZW-115 is induced by decreased expression of TFAM, in addition to suppressing GPX4 and SLC7A11, and increased lipid peroxidation, subsequently, triggers ferroptosis in MiaPaCa-2 cells [101,102,103].

Apart from GEM resistance, the sustained exposure to routine drugs results in tolerance, which, in turn, increases therapeutic doses. As such, dihydroartemisinin (DHA) has the potential to minimize cisplatin (DDP) dosage and maximize the cytotoxicity of cisplatin in PDAC to eliminate cisplatin tolerance. The combination interrupts mitochondrial hemostasis and, consequently, induces ferroptosis by increasing mitochondrial ROS production and the degradation of GPX4 and FTH [109].

In terms of combination therapies, immunotherapy is one of the most prominent ones. Combined treatment with XL888 (HSP90 inhibitor) and anti-PD-1 (checkpoint inhibitor) was shown to be highly effective in a mice model bearing syngeneic subcutaneous (Panc02) or orthotopic (KPC-Luc) tumors, according to Zhang and colleagues [93]. In this study, HSP90 inhibition impacted PSC/CAF in vitro and promoted anti-PD-1 inhibitory efficacy in vivo [93]. The TME can also be influenced by the combination of RSL-3 (ferroptosis inducer) with the novel agents designed by Zhang et al. that disrupt tumor vascular function. By encapsulating RSL-3 in human platelet vesicles (PVs), RSL-3 is improved regarding drug delivery and pharmacokinetics in vivo. This combination also indicates that impairment of tumor vessels can result in tumor embolisms inhibiting nutrient delivery, excessive lipid peroxidation, and mitochondrial dysfunction, resulting in ferroptosis. As a combination therapy to improve the prognosis of PDAC, RSL-3@PVs demonstrated outstanding safety in vitro and in vivo [110]. The research on ferroptosis in conjunction with radiotherapy, primarily for PDAC treatment, is limited. However, radiotherapy with ferroptosis-inducing drugs could provide a new approach to managing advanced and recurrent PDAC [114].

## 3. Necroptosis

An important non-apoptotic immune cell death, necroptosis, exhibits a dual effect on cancer progression. In addition to killing tumor cells, it can promote tumor proliferation, invasion, and metastasis [115,116]. Even though necroptosis is activated when the apoptotic pathway is hindered or suppressed, it is physically and chemically similar to necrosis and apoptosis. Membrane rupture, the cell and its organelles swelling, and the release of inflammatory mediators are some shared characteristics [117,118,119]. Although necroptosis is similar to necrosis and apoptosis, it differs in its ability to be controlled by signaling pathways, the immune system, and the genome, making it a promising candidate for treating pancreatic cancer [120]. Due to this, addressing the interplay between necroptosis and apoptosis, as well as among the necroptosis-related immune system, the genome, and signaling pathways, may help improve the prognosis and treatment for PDAC.

### 3.1. Signaling Pathway

Necroptosis is triggered by various intracellular and extracellular immunogenetically stimuli binding to the Fas (CD95/APO-1) receptor, TNF receptor 1 (TNFR1), and death receptors TRAIL-R, toll-like receptors (TLRs), and the interferon receptor (IFNR) [119,121,122,123]. As pivotal mediators, TLRs are activated by viral dsRNA and lipopolysaccharide (LPS), linking environmental and gene-related factors with immunology in necroptosis [124,125,126]. In the following steps, receptor-interacting protein kinase 1 (RIPK1), as a key promoting factor, determines the cell fate by promoting necroptosis or cell survival. RIPK1, directly or combined with other factors, triggers intracellular necroptosis complexes, including I, IIa, and IIb. Complex IIb, also known as the necrosomes, plays an ultimate role in the progression of necroptosis before cell phenotype changes. Activated RIPK1 and RIPK3 (as necrosome members) lead to the phosphorylation of mixed lineage kinase domain-like (MLKL), causing the necroptosis cell phenotype, including membrane rupture and the release of DAMPs [127,128,129,130]. ZBP1 is a cytosolic nucleic acid sensor stimulated by sensing Z-nucleic acids, leading to RIPK3 phosphorylation and the subsequent activation of complex IIb under the supervision of FADD/caspase-8 [131,132,133]. Caspase-8 and FADD are critical for the regulation of necroptosis, and their inactivity causes the transcriptional upregulation of ZBP1, a key factor in spontaneously phosphorylating MLKL [134,135] (Figure 2).

The crosstalk between different arms of necroptosis and mitochondria transforms mitochondria into a vital organelle for driving necroptosis. As a crucial product of mitochondria, ROS can promote necroptosis by being produced by stimulated RIPK3, inducing RIPK1 autophosphorylation, and activating complex IIb. The level of ROS can be regulated by various immunological or metabolic factors, including TRADD, oxidative phosphorylation (OXPHOS), and PARP1 [122,128,136]. The dysfunction of the oxidative phosphorylation (OXPHOS) in mitochondria can disrupt ATP production and increase ROS levels, both of which can trigger necroptosis [137,138,139]. In PDAC, Solute Carrier Family 25 Member 4 (SLC25A4), a mitochondrial carrier protein responsible for ATP transport, can be inhibited by Poly (ADP-ribose) polymerase 1 (PARP1), resulting in reduced intracellular ATP levels. The activation of PARP1, a DNA damage-repairing enzyme, occurs because of DNA damage. PARP1 overactivity can also impair mitochondrial OXPHOS, increase ROS production, and increase ATP depletion, promoting necroptosis in PDAC [140,141,142,143,144]. Furthermore, ROS are also involved in PDAC cellular migration under the influence of the oncogenic KRAS mutation through the CCL15/ROS axis [145].

### 3.2. Immunogenetics

#### 3.2.1. Necroptosis-Related Genes (NRGs) as a Risk Model in PDAC Prognosis

There is a complex interplay among the immune system, signaling pathways, and gene expression in PDAC, which holds immense potential for therapeutic interventions and prognostic assessments [61,62,65,66]. For instance, evaluating the function of LPS, a double-edged sword factor in PDAC, superbly shows the crosstalk between these factors. Regarding the signaling pathways, studies showed that LPS accelerates PDAC tumor progression and invasion by modifying the NF-kB pathway [146]. LPS activates the PI3K/AKT/mTOR pathway, an oncogenic driver, in PDAC. LPS also has a dual transcriptional effect on PDAC. LPS causes differential expression of various genes, some of which show increased oncogenicity (*WAC*, *PXN*, *INTS6*, *GFPT1*, *ZNF692*, *SNX6*, *PLCD1*, and *PRUNE*), while others show decreased oncogenicity (*ZFAND5, RPL22, Birc-2, SNRPA, GEMIN4, EIF4E, TUSC2*, and *ADAMTS13*) [147]. Regarding the LPS immunological effect, gut-derived LPS can suppress PDAC tumor growth by remodeling the tumor microenvironment and checkpoint transcription [148]. Thereby, by utilizing databases such as TCGA and Gene Expression Omnibus (GEO), multiple models were constructed to forecast the role of necroptosis-related genes (NRGs) in PDAC samples.

A risk model relevant to necroptosis was created to predict the survival and response to treatment of PDAC patients. The genes in the model were found to be upregulated in PDAC compared to normal tissues, including *MYEOV*, *HDAC4*, *TLDC1*, *PITPNA*, *FNDC3B*, *HMGXB4*, and *BAX*, some of which have potential roles in PDAC [149]. *MYEOV*, an oncogene of PDAC, is evaluated in different studies alone or in combination with other genes. The expression level of *MYEOV* and combined methylation and expression levels of the gene *FOXI2* and *MYEOV* were potential prognostic and therapeutic markers for PDAC [150]. *HDAC4* correlated with the proliferative capacity and metastases of smoking-induced PDAC [151]. TLDC1 can also facilitate PDAC proliferation and migration [152]. *BAX* downregulation decreased gemcitabine sensitivity [65]. Although *BAX* was upregulated in the Tang et al. risk model [149], in another risk model developed by Fang et al., this particular factor was placed in the protective factors group for PDAC [65]. This study collected transcriptomic data on PDAC from TCGA, PACA-AU, and PACA-CA cohorts. Various genes (*SPATA2*, *AIFM1*, *SLC25A4*, *BCL2*, *SPATA2L*, *TYK2*, *SMPD1*, *STAT5B*, *SLC25A6*, *USP21*, *STAT4*, *VPS4A*, *RIPK1*, *PLA2G4C*, *IL33*, *CAMK2B*, *MAPK10*, and *BAX*) were protective factors of PDAC prognosis, while 14 genes (*TNFRSF10B*, *HSP90AA1*, *BIRC3*, *TNFRSF10A*, *CHMP4C*, *CASP8*, *FADD*, *CAPN2*, *GLUD1*, *PYGL*, *BIRC2*, *CAPN1*, *CHMP2B*, and *IFNA13*) were risk factors of prognosis. Consensus clustering analysis identified five necroptosis subtypes for PDAC (C1-C5). These subtypes were found to significantly differ in terms of survival outcomes. The C1 and C2 subtypes demonstrated highly activated tumorigenic pathways, while the C3 subtype exhibited the poorest survival rates [65]. Xie et al. conducted a study in which 22 different NRGs were evaluated regarding gene expression and risk prognosis of PDAC [61]. Four genes (*CAPN*, *CHMP4C*, *PYGB*, and *PLA2G4F*) were upregulated, and 18 genes (*IFNA6*, *IFNA2*, *IFNA13*, *BCL2*, *TNF*, *CYBB*, *FASLG*, *JAK3*, *STAT4*, *TNFAIP3*, *PLA2G4C*, *TLR4*, *NLRP3*, *IFNGR1*, *STAT5A*, *TYK2*, *JAK1*, and *SLC25A6*) were downregulated in PDAC tissues. At the same time, only *CAPN2* had higher mRNA levels in PDAC cell lines compared to the normal pancreatic ductal epithelial cell line HPDE6-c7. All the upregulated genes were deemed high risk, while some downregulated genes were deemed low risk (*BCL2*, *JAK3*, *PLA2G4C*, and *STAT4*). Regarding the interplay between gene expression level and overall survival, the worse overall survival was seen in patients with high *CAPN2* and *CHMP4C* and low expression of *PLA2G4C* and *STAT4* in PDAC tissues [61]. These studies suggested that genes known for their role in apoptosis may also function as NRGs. *BCL2*, *PLA2G4C*, and *STAT4* are considered low-risk NRGs, whereas *CAPN2* and *CHMP4C* are considered high-risk NRGs. Additionally, a study discovered that *BCL2*, *CHMP4C*, *IFNA1*, and *TNFAIP3* were regulators prone to mutations in necroptosis-related processes in PDAC [62].

#### 3.2.2. Correlation between NRGs, Immune System, Metabolism, and Signaling Pathways in PDAC

Widmann et al.’s gene set enrichment analysis identified several downregulated metabolic pathways in PDAC, including the metabolism of fatty acids and cholesterol. Conversely, they observed the upregulation of pathways associated with TNFα/NF-κB signaling and the associated process triggered by stress. In PDAC, overall survival is inversely correlated with expression of genes involved in cholesterol synthesis (*ACAT2*, *DHCR7*, *SQLE*, *FDPS*, and *MSMO1*), as well as phospholipid production, modification, and translocation (*OSBPL5*, *PLBD1*, *PITPNM3*, *LPCAT2*, *LPCAT4*, *PNPLA3*, *CPNE3*, *SLC44A1*, and *PLA2R*). This study also showed the upregulation of TNFα/NF-κB signaling pathways and the associated process triggered by stress [63]. Consistent with previous research, Wu et al.’s findings suggested that most NRGs are involved in tumor-related pathways in PDAC. KEGG analysis revealed that NRGs are primarily engaged in the PI3K-AKT signaling pathway, while the p53 signaling pathway is prominently present in PDAC. This finding suggests that NRGs may have the ability to influence PDAC cell invasion and growth by modulating the p53 signaling pathway. The study found that *ALKBH5*, *HNRNPC*, *WTAP*, and *YTHDC2* were more significantly expressed in the high-risk group, while *CASKIN2*, *TLE2*, *USP20*, *SPRN*, *ARSG*, *MIR106B*, and *MIR98* showed substantial expression in the low-risk group [64].

One of the most important roles of NRG emerges through controlling factors associated with the immune system. In keeping with previous studies indicating the positive role of the presence of immune factors, including various types of T cells [153,154,155] and type II interferon IFN-γ [156] in the prognosis and immunotherapy of PDAC, Xie H et al. evaluated the presence of these factors in their suggested NRGs-based risk model (high-risk genes = *CAPN2* and *CHMP4C*; low-risk genes = *PLA2G4C* and *STAT4*). The investigation supported the correlation between NRGs and Necroptosis-Related Immune Factors, with a higher level of T cell infiltration and type II interferon IFN-γ in low-risk groups [61]. Expression of immune checkpoint genes indicates that *HNRNPC* may act as an oncogene, whereas *METTL14*, *WTAP*, *METTL3*, *ALKBH5*, and *YTHDC2* may act as tumor suppressors [64]. Chemokines, another immune system member, play a crucial role in PDAC progression, prognosis, and immune response by regulating several pathways, including necroptosis. The number of neutrophils, a key prognostic and therapeutic factor in the TME of PDAC, is tightly regulated by factors such as CXC-chemokine receptor 4 (CXCR4) signaling [157]. CXCL5 induces immunosuppressive cell infiltration, including neutrophils, leading to poorer outcomes in PDAC [158]. Necroptosis shows a different face in PDAC by releasing CXCL5 from necroptotic cells and promoting cancer cell migration and invasion via CXCR2, indicating the dual role of necroptosis in PDAC [159]. A recent study demonstrated that the tumor-infiltrating myeloid cells (TIMs) level in PDAC affects clinical outcomes. A scRNA-seq analysis of PDAC patients identified 10 upregulated genes associated with necroptosis in PDAC tumors and 5 upregulated genes in the surrounding area of the tumor and selected blood samples. Additionally, different myeloid cell sub-clusters had different prognostic clinical values in PDAC [160].

The interplay between the genome and TME in cancer cell proliferation, survival, and resistance to therapy was evaluated in the study conducted by Lu et al. In this study, a system was designed that combines necroptosis and immunity to predict the TME and treatment targets in PDAC. The necroptosis-immune (NI) score showed predictive competence for chemotherapy and immunotherapy. In this study, the high necroptosis–highimmunity (HNHI) group had the best prognosis, while the low necroptosis–low immunity (LNLI) group had the shortest survival time. The prognoses of the LNHI and HNLI groups were between those of the above two phenotypes. In this study, *SLC2A1* was a key component of the NI score, and an oncogene correlated to necroptosis and the TME in PDAC [66]. According to previous studies regarding the SLC group, *SLC2A1* [66] and *SLC44A1* [63] were oncogenes, while *SLC25A6* and *SLC25A4* [65] had protective roles.

### 3.3. Treatment

#### 3.3.1. Drugs That Modulate Necroptosis Signaling Pathways

In order to effectively treat PDAC, resistance to apoptosis, a natural process of programmed cell death, must be overcome. Therefore, it would be worthwhile to investigate necroptosis as an alternative to inducing cell death in pancreatic cancer cells. Inhibiting pancreatic cancer cell invasion and migration with necroptosis can be accomplished by modulating the immune system, such as SB225002 (Table 4), which inhibits CXCR2 [159]. Other drugs may cause necroptosis in pancreatic cancer cells by modulating signaling pathways, primarily necrosome formation, and organelle dysfunction. In this way, multiple cell death is induced due to common factors.

The utilization of light to destroy specific tissue by stimulating a photosensitizer (PS), which is given intravenously and accumulates in the tumor before being irradiated, is known as photodynamic therapy (PDT). Selectivity is attained by focusing light on the desired tissue through various techniques for light delivery. When PDAC cells are unresectable and have spread to adjacent tissue, PDT can be considered a suitable treatment option [161,162]. The utilization of methylene blue (MB) as a PS in PDT demonstrated increased expression of RIPK1, RIPK3, and MLKL in transformed cells. By activating necroptosis, photodynamic therapy with methylene blue (PDT-MB) can be a valuable addition to treating PDAC, reducing local and metastatic recurrences and microscopic residual disease [163]. PDT can enhance treatment efficacy by utilizing different irradiation regimens and a specific PS to induce diverse cell death responses (apoptosis, parthanatos, mitotic catastrophe, pyroptosis, necroptosis, and ferroptosis). Nevertheless, for optimal therapeutic outcomes, PDT must be integrated with other modes of cell death and strive to stimulate immunogenic cell death pathways, which can enhance the patient’s overall survival and quality of life [164].

Silver nanoparticles (AgNPs) trigger mixed cell death in PDAC, including apoptosis, autophagy, necrosis, necroptosis, and mitotic catastrophe. They achieve this by altering cell or organelle features, signaling pathways, and protein expression [165,166]. AgNPs also disrupt the antioxidant (SOD1, SOD2, GPX-4, CAT, and SOD3) system in PDAC cells and induce oxidative and nitro-oxidative mechanisms [167]. Regarding necroptosis, AgNPs alter two NRG expression levels, increasing BAX and decreasing BCL-2 expression. Additionally, AgNPs play a necroptosis-induced role by increasing RIP-1, RIP-3, and MLKL levels. Pancreatic cancer cells are more susceptible to AgNPs-induced cytotoxicity than non-tumor cells. Furthermore, AgNPs’ cytotoxic effects on pancreatic cancer cells are size- and concentration-dependent [165,166].

The induction of mixed cell death by IMB5036, a novel pyridazinone compound, is an effective treatment for pancreatic cancer. Upon exposure to this compound, pancreatic cancer cells exhibit various changes, including membrane blebbing (a characteristic of apoptosis and pyroptosis) and swelling (a characteristic of necroptosis). Despite IMB5036′s ability to partially activate apoptosis and pyroptosis, its primary mode of action is necroptosis, initiated by the upregulation of RIPK1, RIPK3, and largely MLKL in pancreatic cancer cells [168].

The overexpression of a serine/threonine kinase, called AURA kinase (AURKA), a negative regulator of necroptosis and apoptosis, correlates with worse overall survival in PDAC patients [169,170,171]. As AURKA prevents RIPK3–MLKL activation in PDAC cells, AURKA inhibitors may be a promising treatment for PDAC [171]. MLN8237 (alisertib), an AURKA inhibitor, suppresses pancreatic cancer cell proliferation and cellular migration by promoting various cell death, including apoptosis and necroptosis [172,173]. A highly selective AURKA inhibitor promoting different cell death activities, CCT137690, induces necrosome formation via RIPK1, RIPK3, and MLKL in PANC-1 PDAC cells [171].

#### 3.3.2. Drugs That Modulate Organelle Function

Several agents for treating PDAC affect necrosome formation and the changing intrinsic systems of organelles, such as ROS accumulation, mitochondrial function, and ER function. In this manner, some drugs that are effective in treating diabetes produce positive results since there is a connection between obesity, pancreatic cancer progression, and pancreatic-cancer-related diabetes. PDAC-related diabetes can be differentiated from type II diabetes through the measurement of both Adiponectin (a hormone secreted by adipocytes) and the interleukin-1 receptor antagonist (IL-1Ra) [180]. Adiponectin (APN) is an anti-tumor and anti-angiogenesis agent that inversely correlates with pancreatic cancer [174,175]. As a result of increasing mitochondrial Ca^2+^ levels, which produce superoxide and, subsequently, activate RIPK1, pancreatic cancer cells undergo necroptosis in response to agonists of the APN receptor (AdipoRon). In addition, Akimoto et al. demonstrated that when MIAPaCa-2 cells are injected into nude mice, oral administration of AdipoRon diminishes proliferation and angiogenic activity in PDAC [174]. Obesity-associated factors, particularly leptin, reduced the anticancer activity of AdipoRon. Human pancreatic cancer also exhibited chemoresistance to anticancer drugs due to obesity and leptin signaling [175]. In addition, another anti-diabetic drug, the Vanadium compound, triggered various cell death in pancreatic cancer by inhibiting the cell cycle, increasing ROS, and upregulating RIPK1 and RIPK3 in a dose-dependent manner. Quinolones and phenanthrolines, as vanadium compounds with organic ligands, induced necroptosis in PANC-1 cells [176].

#### 3.3.3. Combination Therapy

In combination therapy, the lower dose of some drugs can be used to achieve a better outcome and overcome resistance to treatment. The efficacy of combination therapy with PDT-MB was previously discussed; however, electrochemotherapy (ECT) is another physical treatment that can enhance the effectiveness of chemical therapies. ECT can effectively treat pancreatic cancer due to its ability to enhance the chemotherapy response and reduce drug resistance. Through ECT, cell membrane pores are enlarged, allowing for better drug absorption and minimizing the damage to healthy tissues. Combining ECT with bleomycin, cisplatin, and oxaliplatin results in necroptosis rather than apoptosis in tumor cells [181,182]. Combination therapy proved effective in overcoming drug resistance caused by apoptosis resistance, particularly in inducing necroptosis in pancreatic cancer. A notable example is gemcitabine (which predominantly induces apoptosis) combined with necroptosis-inducing drugs. In some cases, PDAC patients positively responded to the AURKA inhibitor alisertib (MLN8237), when safely used alongside gemcitabine [177]. It was reported that SK, a naphthoquinone derivative, reduced the size of PANC-1 tumors and induced necroptosis. The tumor volume further decreased when combined with gemcitabine, in support of SK enhancing the anti-tumor effects of gemcitabine [178]. PDAC patients may also benefit from targeting liver X receptors (LXR), which impair cholesterol and the phospholipid metabolism. A small molecule LXR modulator, GAC0003A4 (3A4), inhibits the expression of LXR downstream genes and pathways in PDAC cells to induce necroptosis and apoptosis. In addition to the efficacy of 3A4, combinations of 3A4 and gemcitabine may enhance their cytotoxic effect [63].

Hannes et al. conducted a study that focused on modulating the immune system to induce necroptosis, in line with the proven effectiveness of combination therapy. In this study, the researchers demonstrated a combination of a drug called BV6, which mimics a protein called Smac, and a ligand called 2′,3′-cyclic guanosine monophosphate–adenosine monophosphate (2′3′-cGAMP), which activates a protein called STING, to induce necroptosis in pancreatic cancer cells. The researchers found that the combination of BV6 and 2′3′-cGAMP effectively triggered necroptosis in pancreatic cancer through MLKL phosphorylation and the stimulation of NF-κB, type I interferons (IFNs), TNFα, and IFN-regulatory factor 1 (IRF1) signaling pathways [179]. Interestingly, as a key member of the kindlin family, fermitin family member 1 (FERMT1) was identified as a promising diagnostic and prognostic indicator for PDAC. FERMT1 is associated with immune cell infiltration and regulates m6A and necroptosis. The positive correlation between *FERMT1* and the three main genes responsible for necroptosis (*RIPK1, RIPK3*, and *MLKL*) indicates that *FERMT1* may serve as NRGs in PDAC. It was shown that individuals with elevated levels of *FERMT1* respond more to Palbociclib (CDK4/6 inhibitor), TAE-226 (focal adhesion kinase selective inhibitor), and AM-5992 (AMG-925; dual inhibitor of CDK4 and FLT3), which can be combined in combination therapy to enhance the treatment of pancreatic cancer [183]. There is evidence that stimulating necroptosis can be a successful supplementary treatment for pancreatic cancer.

## 4. Pyroptosis

Pyroptosis, a caspase-driven non-apoptotic necrotic cell death, is accompanied by morphological alterations, such as swelling, bubble-like protrusions, and membrane ruptures, leading to the secretion of inflammatory mediators [184,185]. There are two types of cell death, lytic and non-lytic, based on the morphological changes and rupture of the membrane. As opposed to apoptosis, which is a non-lytic caspase-driven cell death that does not release pro-inflammatory factors, pyroptosis is a caspase-driven cell death that, like necroptosis, leads to lytic cell death, producing highly inflammatory factors [30,31,33]. Although the induction of pyroptosis was considered a promising therapeutic strategy, the release of pro-inflammatory factors was associated with tumorigenesis and drug resistance [186,187]. Pyroptotic-related genes (PRGs) can control the TME, the prognosis, and pancreatic cancer progression [78]. An essential immune system component, NLRP3 (NOD-like receptor family pyrin domain-containing protein 3), is a pyroptosis-inducer. The upregulation of NLRP3 can either be pro-tumor (colorectal cancer) [188] or a tumor suppressor (hepatocellular cancer) [189], indicating the conversation among signaling pathways, immunity, and genomes in cancer. Additionally, in the context of crosstalk between the genomes and signaling pathways of pyroptosis in different types of cancer, the modulation of the pyroptosis-activator gasdermins (*GSDM*) gene has different effects on tumor development [190,191]. If *GSDM* is knocked out in lung cancer, it may act as a tumor suppressor [190]. However, if *GSDM* is silenced in gastric cancer, it may be a pro-tumor factor [191]. As a result, pyroptosis is a double-edged sword in many cancers [186,187]. In order to develop effective therapies for pancreatic cancer and improve survival rates, it is essential to explore the different aspects of the interplay between pyroptosis and signaling pathways, the immune system, and genomes.

### 4.1. Signaling Pathway

Signaling pathways involved in pyroptosis include both canonical and non-canonical pathways. Damage- or pathogen-associated molecular patterns (DAMPs or PAMPs) are the primary initiators of the canonical pathway by stimulating the pattern recognition receptors (PRRs). NLRP3, as a member of PRRs, an apoptosis-associated speck-like protein containing a CARD (ASC), and procaspase-1 form the inflammasome [192,193,194]. By activating caspase-1, GSDM, the main executor of pyroptosis, is cleaved. The cleavage of GSDM leads to membrane pores formation following the separation of the N-terminal pore-forming segment from the C-terminal repressor segment, thereby releasing inflammatory mediators [193,194]. IL-1β and IL-18 are two inflammatory mediators activated by activated caspase-1; their release is a key characteristic of pyroptosis [33,194,195,196]. Similarly, the activation of GSDM by other caspase family members (caspase-4/-5/-11) in a non-canonical pathway also results in the same pyroptotic phenotype. Bacterial LPS is the main activator of the non-canonical caspase family [33,196,197,198]. Furthermore, caspase-1, as the main driver of the canonical pathway, can be independently activated through Mammalian STE20-like kinase 1 (MST1) activation. As a result, PDAC can be suppressed by MST1, a member of the Hippo signaling pathway, through ROS-induced pyroptosis [68] (Figure 3).

### 4.2. Immunogenetics

#### 4.2.1. Pyroptosis-Related Genes (PRGs) Risk Model in Pancreatic Cancer Prognosis

Even though our knowledge of the genome concerning pancreatic cancer is still incomplete, PRGs can be considered a positive aspect of treating pancreatic cancer. Several studies used PRGs to study different human gene datasets, including the TCGA cohort, using different analysis methods, such as the most minor absolute shrinkage and selection operator (LASSO) method, which results in the development of different risk models. It was demonstrated that *MST1*, a tumor suppressor, is underexpressed in PDAC cells, which have a prominent role in tumor progression [68]. Notably, PDAC patients with different pathological stages were found to have considerably varied expressions of inflammasome-related genes [199]. Overexpression of two subtypes of *GSDM, GSDME* and *GSDMC*, positively correlated with poor prognosis and chemoresistance in PDAC [69,70].

Tao et al. developed a risk model via five PRGs (*ELANE*, *GSDMC*, *IL18*, *NLRP1*, and *NLRP2*). They demonstrated that the expression level of most PRGs substantially differs between normal pancreatic tissue and PDAC, indicating an observable alteration in the pyroptotic function, either through an increase in the copy number or demethylation. Higher neutrophil-derived active neutrophil elastase (*ELANE*) and *NLRP1* expression predicted an improved prognosis among these core genes. In this risk model, poor prognosis was also associated with overexpression of *GSDMC*, *IL-18*, and *NLRP2* in patients with PDAC [72]. Another study used *IL-8* and *GSMDC* alongside *PLCG1* and *AIM2* to design a PRG risk model, in which the expression levels of *IL-18*, *GSDMC*, *AIM2*, and *GSDMC* were higher in most of the human PDAC cell lines compared to the hTERT-HPNE cell line [71]. It was shown in both previous studies that low-risk populations respond better to immunotherapy and chemotherapy and have better overall survival. This finding emphasizes the importance of genomes in cancer progression and prognosis.

#### 4.2.2. Correlation among PRGs, Immune System, Metabolism, and Signaling Pathways in Pancreatic Cancer

Functional enrichment analysis of PRGs revealed that PRGs were mainly involved in pyroptosis, apoptosis, and other immune signaling pathways (TNF, TLR, and inflammatory response), indicating a conversation among the genome, signaling pathways, and the immune system to connect apoptotic cell death with pyroptosis [77].

The seven pyroptosis-related lncRNAs (PRlncRNAs), consisting of *AC083841.1*, *AC090114.2*, *AC005332.6*, *PAN3-AS1*, *LINC01133*, *AC087501.4*, and *AC015660.1*, were used to establish a risk signature in which *AC083841.1*, *LINC01133*, and *AC015660.1* were categorized as potentially compromising lncRNAs. However, the remaining ones were considered protective. This study’s risk score directly correlated with poor overall survival and diminished immunity. Since the high-risk group had reduced immune infiltration, it did not benefit from immune therapy as much as the low-risk group with a higher level of immune infiltration. Accordingly, the prognosis, therapeutic options, the TME design, and immune cell enrichment are interconnected in pancreatic cancer [73]. Additionally, the high-risk group had more tumor mutations [73], consistent with the study conducted by Xu and colleagues using different PRGs to develop risk models [74]. According to this study, patients with a lower tumor mutation in the low-risk group had an improved survival rate. Significant differences exist between the high-risk and low-risk groups in differentially expressed genes engaged in immune-response pathways, suggesting that pyroptosis may affect the TME of PDAC. In this PRG risk model, the poor-prognostic high-risk group had poor immunity and an increased tumor purity in the TME. The better survival in this study was associated with the upregulation of *APIP*, *CHMP6*, and *PLCG1* and the downregulation of *AIM2*, *CASP4*, *CASP6*, *CHMP4C*, *GSDMC*, and *GZMB* [74]. Almost similar results were obtained in another study that utilized different genes. A PRGs prognostic index (PRGPI) was developed utilizing eight PRGs, of which *AIM2*, *GBP1*, *HMGB1*, *IL18*, *IRF6*, and *NEK7* exhibited risk effects, while *NLRP1* and *PLCG1* exhibited protective effects in this study. The PRGs in this risk model confirmed the previous study results, which showed that the risk score had an indirect association with better overall survival, a direct association with mutation (including oncogene *KRAS* and tumor suppressor genes involving *TP53*, *SMAD4*, and *CDKN2A*), and the potential to display immune suppressed characteristics. Accordingly, high-risk groups exhibited enriched plasma cells and M1 macrophages and overexpression of immune checkpoints and HLA family genes (such as *PD-L1*) [75].

A significant increase in most m6A-related genes (except *HNRNPC*) was observed in the low-risk group of the Li et al. risk model, suggesting that pyroptosis may be related to m6A modification. According to this study, PRGs risk models were defined with the combination of immune and signaling pathways genes (*CASP4*, *GSDMC*, *NLRP1*, *PLCG1*, *IL-18*, *CASP1*, and *NLRP2*), among which *CASP4*, *NLRP1*, *PLCG1*, *IL-18*, and *CASP1* are overexpressed in pancreatic cancers. More characteristics in the low-risk group indicate that the patients in this group tend to respond more to immunotherapy than patients in the high-risk group whose chemotherapy (which includes rapamycin, paclitaxel, and erlotinib) is more effective. The low-risk group in this study showed higher levels of CD8+ T cells, immune and stroma scores, and immune checkpoint expression (notably CTLA4 and PD-1) [76]. An additional risk model PRG based on more immune-related genes (*CASP4*, *GSDMC*, *IL-18*, *NLRP1*, *NLRP2*, *PLCG1*, *TIRAP*, and *TNF*) was developed, which can be used to predict pancreatic cancer prognosis with an accuracy of medium to high. As described by Li et al., this study revealed that *NLRP1*, *NLRP2*, *IL18*, and *CASP4* overexpression may contribute to a lower overall survival rate in pancreatic cancer. In this study, *TNF* was also underexpressed in pancreatic cancer cells [77]. Another study employing almost different PRGs in developing the risk model revealed that the high-risk group decreased anti-tumor immunity by impairing CD8 T and NK cell infiltration in the TME of pancreatic cancer. In addition to *CASP4*, *GSDMC*, *NLRP1*, *PLCG1*, and *IL18*, this PRGs risk model also includes other immune- and signaling-related genes, including *TLR3*, *IFR1*, and *GPX4*. Interestingly, the upregulation of *TLR3* was shown to facilitate the proliferation, migration, and invasion of pancreatic cancer cells [78].

Pyroptosis and apoptosis are likely connected, as a knockdown of GSDME converted pyroptosis into apoptosis in PDAC [69]. In light of the prognosis-related PRGs that Xu and colleagues designed, there is a correlation between PRGs and cancer progression by modulating the immunity and signaling pathways in pyroptosis and apoptosis. As a result, five highly expressed PRGs, including apoptotic-related genes (*BAK1* and *TP63*), *CHMP4C*, *IL18*, and *NLRP2*, were categorized as high risk in this prognostic model and provided poor prognoses for pancreatic cancer. Interestingly, the high-risk group exhibited fewer invading immune cells and the reduced activation of immune-related pathways [79]. Consequently, there is a relationship between different types of cell death, and it is difficult to distinguish the separate roles played by each in shaping the TME. Activating the genes related to one type of cell death may affect other types, providing pancreatic cancer patients an opportunity to improve treatment efficacy. In line with this, Yu and colleagues developed a pyroptosis–ferroptosis (P-F) score for PDAC, in which lower P-F scores were associated with a more immune-suppressed phenotype, increased genomic mutations, impaired immunotherapy responses, and the prognosis. Regarding gene modification, the genes *KRAS*, *TP53*, *SMAD4*, and *RNF43* demonstrated substantial co-expression for low P-F scores, in which ferroptosis plays a major role [200]. Yu et al. also stratified PDAC into four cell death subtypes, quiescent, pyroptosis, ferroptosis, and mixed, based on the expression profile of pyroptosis- and ferroptosis-related genes [200]. Tumors with co-expressed upregulation of pyroptosis- and ferroptosis-related genes (the mixed subtype) exhibited an adverse prognosis. Regarding the TME, the pyroptosis subtype contained more activated T cells (CD4+ and CD8+), whereas the mixed subtype contained the least number of activated NK cells. Zuo and colleagues also revealed that a better prognosis for PDAC patients is associated with higher T cells, NK cells, and macrophages, which enhance cytolytic and inflammation levels. According to this study, a risk score was based on genes involved in signaling pathways, immune systems, and glycolysis as a metabolic pathway [199]. Additionally, overexpression of *GSDMC*, a poor prognostic factor, correlated with *PD-L1* overexpression and poor infiltration of CD8+ T cells in the TME of PDAC models [70].

Regarding evaluating crosstalk between genomes, metabolism, and signaling pathways via pyroptosis, a prognostic PRGs model based on *CASP4* and *NLRP1* was developed. CASP4 may contribute to PDAC progression through tumor growth and migration and is identified as a key regulator of the PDAC lipid metabolism by this study. The knockdown of *CASP4* significantly reduced the number of lipid droplets in PANC-1 and AsPC-1 cells, in addition to downregulating fatty acid synthesis mediators. In contrast, by modulating MAPK, mTOR, and JAK/STAT signaling pathways, NLRP1, which is inversely correlated with KRAS, may serve as a tumor suppressor for PDAC. It should be noted that trametinib’s efficacy showed direct and inverse correlations with the expression levels of *CASP4* and *NLPR1*, respectively [67]. By upregulating enzymes that participate in the sphingolipid metabolic pathway, chemoresistance can be alleviated in pancreatic cancer animal models [201]. Thus, targeting metabolic pathways may be a promising therapeutic avenue requiring further investigation.

### 4.3. Treatment

#### 4.3.1. Drugs That Modulate Signaling Pathways and Other Cell Death through Pyroptosis

As previously mentioned, the canonical pathway of pyroptosis comprises three main arms: MST1, ROS, and caspase-1. In PDAC, caspase-1 inhibition or ROS removal induces pyroptosis mediated by MST1. VX-765 (caspase-1 inhibitor) and N-acetyl-cysteine (NAC) (ROS scavenger) use this method to inhibit PDAC cell proliferation, migration, and invasion via MST1 (Table 5) [68]. Reduced ROS production due to overexpression of the sphingolipid metabolic enzyme (ASAH2), preventing chemotherapy-induced pyroptosis, in turn, causes chemoresistance. The Src-signal transducer initiates this cascade, leading to a decrease in ceramide levels. In vivo and in vitro, increasing ceramide levels by ASAH2 or the Src inhibitor, such as ceramidase inhibitor B13 or dasatinib, improved chemosensitivity in pancreatic cancer [201].

#### 4.3.2. Combination Therapy

Metallodrugs showed the potential to improve pancreatic cancer treatment in various ways, including the ability to pass the dense TME, improve photosensitivity, and induce the co-occurrence of different cell death pathways [165,166,167]. An innovative platinum-based metallodrug complex, SEP (quinone derivative seratrodast (STD) plus cisplatin (CDDP)), was developed to eliminate apoptosis-resistant PDAC. In addition to efficiently penetrating cancer cells and damaging DNA, SEP also caused mitochondrial superoxide anion radical formation and the subsequent triggering of pyroptosis, necroptosis, and apoptosis [202]. The process is known as PANoptosis, and it plays a significant role in developing strong anti-tumor immunity by secreting inflammatory mediators [203]. The evidence showed that SEP more efficiently suppresses *KRAS*-mutant PDAC cells than CDDP, supporting the potential for combination therapy [202].

Although overexpression of *GSDME* was associated with poor prognoses in PDAC, its high expression level and ability to promote apoptosis and pyroptosis make it a potential therapeutic target. In response to PDAC chemotherapy (gemcitabine, irinotecan, 5-fluorouracil, paclitaxel, and cisplatin), pyroptosis and apoptosis were concurrently induced through the cleavage of GSDME by caspase-3, followed by the activation of pro-apoptotic caspase-7/8 [69]. Furthermore, steroidal saponins PPI/CCRIS/PSV (polyphyllin I (PPI), collettiside III (CCRIS), and paris saponin V (PSV)) induce the caspase-3-mediated cleavage of GSDME, potentially suppressing PANC-1, AsPC-1, and BxPC-3 PDAC cell growth [204]. By activating the caspase-3-mediated cleavage of GSDME, the sonodynamic–immunomodulatory pyroptotic strategy provides a significant immune response against tumors. LY364947 (a TGF-β receptor inhibitor), the sonosensitizer, produced ROS in response to ultrasound exposure and caused GSDME to be cleaved by caspase-3 and broke the dense TME via collagen degradation, resulting in T-cell infiltration and almost complete tumor eradication in mouse models [205]. Metabolic sonosensitizers in sonodynamic therapy (SDT) also demonstrated a significant effect on pancreatic cancer treatment. Two metabolic sonosensitizers were developed by Yang and collaborators by combining 5-aminolevulinic acid hydrochloride (ALA), a lipid, and poly(lactic-coglycolic acid) (PLGA) microbubbles (MBs) [206]. These compounds were designated ALA-lipid MBs and ALA-PLGA MBs, respectively. The combination of ALA-lipid/PLGA MBs-mediated SDT produced significantly more ROS than ALA-mediated SDT, which resulted in apoptosis and pyroptosis co-occurring and an enhanced response to therapy in AsPC-1 and BxPC-3 cells [206].

In addition to SDT, PDT converts the immunosuppressive cold TME to the immunogenic hot TME in pancreatic cancer by inducing an immune response through pyroptosis [203,207]. Combined with PDT, TBD-3C (a membrane-targeted photosensitizer) stimulates pyroptosis and the subsequent hot TME due to M1 polarization and the maturation and activation of dendritic cells and CD8+ T cells in response to TBD-3C-induced pyroptotic cells [207].

## 5. Conclusions and Perspectives

This review emphasizes the importance of non-apoptotic cell death pathways, such as ferroptosis, necroptosis, and pyroptosis, in prognosis and drug resistance in PDAC. These pathways can work independently of, synergistically with, or in conjunction with apoptosis to determine the fate of PDAC cells. Moreover, our review highlights the impact of immunogenetic factors on ferroptosis, necroptosis, and pyroptosis, providing valuable insights into their implications for PDAC treatment and prognosis. Future studies can focus on developing a clinically effective risk model that incorporates a non-apoptotic gene-related (NAGR) prognostic risk model, including ferroptosis-related genes (FRGs), necroptosis-related genes (NRGs), and pyroptosis-related genes (PRGs), to further elucidate the role of these pathways in immunity, metabolic regulation, and signaling cascades. Ultimately, this knowledge can contribute to improving prognoses and developing effective treatment strategies for PDAC. Further, it is noteworthy that genomes, immunity, and metabolic pathways contribute, individually and in combination, to various aspects of non-apoptotic cell death biology in PDAC. These mechanisms play a critical role in non-apoptotic cell death signaling and crosstalk with other pathways. Future studies will be required to determine how to apply this knowledge to the clinical management of PDAC patients.

## Figures and Tables

**Figure 1 biomedicines-11-02792-f001:**
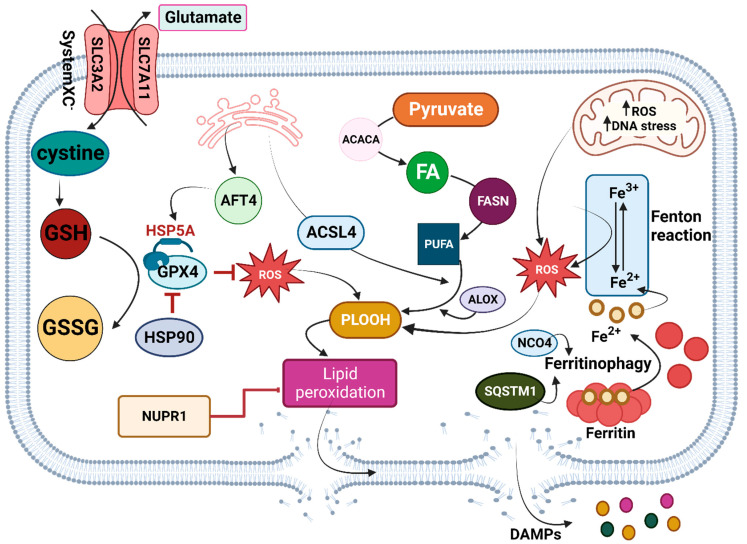
Signaling pathways involved in ferroptosis. After entering, cystine via system XC^−^ antiporter GPX4 works as an antioxidant and reduces intracellular ROS. However, HSPA5 enhances GPX4 antioxidant activity by stabilizing this enzyme. HSP90 inhibits this antioxidant and subsequently increases ROS. On the one hand, ER enhances GPX4 via AFT4, facilitating HSPA5 expression. On the other hand, it facilitates PLOOH by releasing ACSL4. Mitochondria increases ROS levels via different cycles, including the Fenton reaction and TCA. Iron accumulation, the product of ferritinophagy or degradation of SLC40A1, provides an ingredient for the Fenton reaction. Ferritinophagy is also enhanced by autophagy receptors, NCO4, or SQSTM1. Lipid peroxidation results in ferroptosis and pore formation, which NUPR1 can inhibit. Abbreviations: SLC32A, Solute Carrier Family 32; ASLC7A11, Solute Carrier Family 7A11; AFT4, Adaptive Fourier Transform 4; GSH, glutathione; GSSG, oxidized glutathione; GPX4, glutathione peroxidase 4; HSP90, Heat Shock Protein 90; HSP5A, Heat Shock Protein 5A; ROS, Reactive Oxygen Species; NUPR1, Nuclear Protein 1; ACSL4, Acyl-CoA Synthetase Long-Chain Family Member 4; ACACA, Acetyl-CoA Carboxylase Alpha; FA, fatty acid; FASN, fatty acid synthase; PUFA, Polyunsaturated Fatty Acid; ALOX, Arachidonate Lipoxygenase; PLOOH, phospholipid hydroperoxide; NCO4, nuclear coactivator 4; SQSTM1, sequestosome 1. Created with BioRender.com. Accessed on 30 July 2023.

**Figure 2 biomedicines-11-02792-f002:**
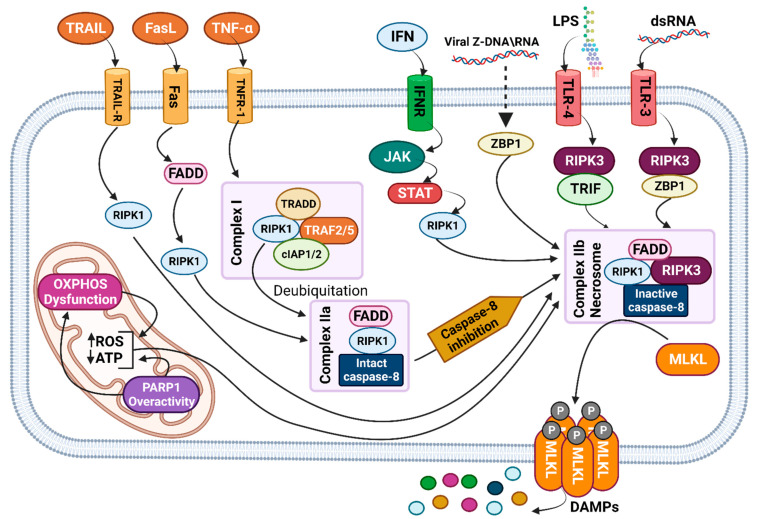
Necroptosis signaling pathways. Binding death receptor family (Fas receptor, TNFR1, and TRAIL-R) and other immunological receptors (TLRs and IFNR) to their ligands leads to the activation of initiator factors. Complex I members (TRADD, TRAF2/5, cIAP1/2, and RIPK1) are activated by TNFR-1 stimulation. This process inhibits the formation of complex II groups, including complex IIa (caspase-8, FADD, and RIPK1) and complex IIb (caspase-8, FADD, RIPK1, RIPK3, and MLKL). RIPK1 deubiquitylation leads to the formation of complex II groups, promoting downstream events. So cell fate depends on how RIPK1 changes, which can also be activated independently of complex I. Activating RIPK1 by Fas, TRAIL-R, INFR, and ZBP 1 stimulates complex IIa and IIb independently of complex I. Additionally, extrinsic factors like LPS stimulate TLR4, directly resulting in the activation of complex IIb. OXPHOS dysfunction or PARP1 overactivity by reducing ATP and increasing ROS stimulates complex IIb. Activation of RIPK3 by inactivated caspase-8 results in phosphorylation and oligomerization of MLKL, which translocates to the cell membrane, causing membrane rupture and release of DAMPs. Abbreviations: TNFR1, Tumor Necrosis Factor Receptor 1; TRAIL-R, TNF-Related Apoptosis-Inducing Ligand Receptor; TLRs, toll-like receptors; IFNR, interferon receptor; TRADD, TNFR1-Associated Death Domain; TRAF2/5, TNF Receptor-Associated Factor 2/5; cIAP1/2, Cellular Inhibitor of Apoptosis Protein 1/2; RIPK1, Receptor-Interacting Protein Kinase 1; MLKL, mixed lineage kinase domain-like; LPS, lipopolysaccharide; OXPHOS, oxidative phosphorylation; PARP1, Poly(ADP-ribose) Polymerase 1; ATP, Adenosine Triphosphate; ROS, Reactive Oxygen Species; ZBP1, Z-DNA Binding Protein 1. Created with BioRender.com. Accessed on 30 July 2023.

**Figure 3 biomedicines-11-02792-f003:**
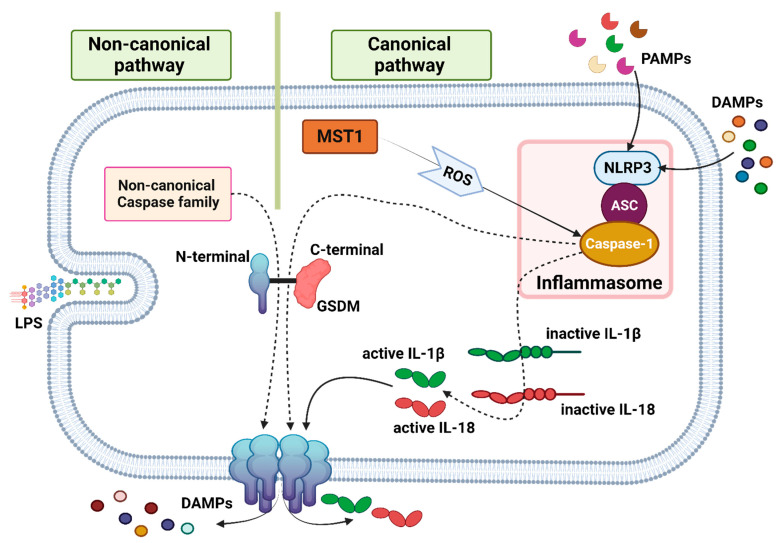
Pyroptosis signaling pathways. Activation of every member of the non-canonical caspase family (caspase-4/-5/-11) via bacterial LPS initiates the non-canonical pyroptosis signaling pathway. The canonical signaling pathway is triggered by the activation of NLRP3 via PAMPs or DAMPs or by directly activating caspase1 via the MST1 byproduct (ROS), the main functional component of the inflammasome group. GSDM, the main executor of pyroptosis, is cleaved both by a canonical and non-canonical pathway, leading to the activation of the N-terminal domain, which results in the formation of pores. The other products of the canonical pathway are activated IL-1β and IL-18, which are released from the cell with DAMPs via GSMD-induced pores. Abbreviations: LPS, lipopolysaccharide; PAMPs, pathogen-associated molecular patterns; DAMPs, damage-associated molecular patterns; NLRP3, NOD-like Receptor Family Pyrin Domain Containing 3; ROS, Reactive Oxygen Species; GSDM, gasdermin; IL-1β, Interleukin-1 Beta; IL-18, Interleukin-18; MST1, Mammalian Sterile 20-Like Kinase 1. Created with BioRender.com. Accessed on 30 July 2023.

**Table 1 biomedicines-11-02792-t001:** Apoptotic and non-apoptotic cell death.

Cell Death	Immunogenic Feature	Lytic Feature	Major Morphological Change	Involved Organelles	Pore Executer	Caspase-Dependent	Main Signaling Pathway	Refs.
Apoptosis 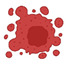	Generally non-immunogenic. Apoptotic cells are typically eliminated by phagocytosis without initiating a strong immune response.	Non-lytic	Cell shrinkage, intact cell membrane, membrane bubbling	Mitochondria	_	Dependent (caspase-3/-7/-8)	Apoptotic bodies are formed when caspase -3/7 is activated during extrinsic and intrinsic pathways. Extrinsic pathways activate caspase-3/-7 via activated caspase-8, and intrinsic pathways activate caspase-3/-7 via Bax/Bak-induced mitochondrial DNA fragmentation.	[27,28,29,30,31,32]
Ferroptosis 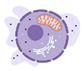	Moderately immunogenic. Can release DAMPs.	Not classically lytic, nanopores in the membrane may not disrupt the membrane integrity to the same extent as necroptosis or pyroptosis.	Cell swelling, nanopore	Mitochondria	Lipid peroxidation	Independent	Inhibition of XC-/GSH/GPX4 induces lipid peroxidation and iron accumulation.	[27,28,29,30,31]
ER
Necroptosis 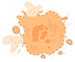	Highly immunogenic because of the production of abundant DAMPs. Activates immune cells and promotes inflammation.	Lytic, characterized by rupture of the cell membrane and release of intracellular contents.	Cell swelling, pore formation, plasma membrane bubbling	Mitochondria	P-MLKL	Independent	Activation of initiator factors by the death receptor family leads to necrosome formation, MLKL phosphorylation, and DAMP release.	[27,28,29,30,31,33]
ER
Pyroptosis 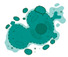	Highly immunogenic. A strong inflammatory response is triggered due to the release of cytokines (IL-1β and IL-18). Promotes adaptive immune system response.	Lytic, the rupture of a cell membrane caused by GSDM releasing cytokines and cell contents.	Cell swelling, pore formation, plasma membrane bubbling	_	N-terminal GSDM	Dependent	Canonical or non-canonical caspases cleave and activate GSDM to form pores.	[27,28,29,30,31,33]
(caspase-1 (canonical pathway) and caspase-4/-5/-11 (non-canonical pathway))

Abbreviation: ER, endoplasmic reticulum; GSH, glutathione; GPX4, glutathione peroxidase 4; MLKL, mixed lineage kinase domain-like; GSDM, gasdermin. Created with BioRender.com. Accessed on 30 July 2023.

**Table 2 biomedicines-11-02792-t002:** Implications of important non-apoptotic cell-death-related genes.

Type of Cell-Death-Related Genes	Upregulated Genes Correlated with Better Prognosis	Upregulated Genes Correlated with Adverse Prognosis	Ref.
FRGs	*CAV1*, *DDIT4*, *SRXN1*, *TFAP2C*, *MT1G*, *TUBE1*, *ATG4D*, *ENPP2*, *SETBP1-DT*, *ZNF93-AS1*, *SLC25A5-AS1*, *AC073896.2*, *LINC00242*, *PXN-AS1*, *AC036176.1*	*SLC40A1*, *PTGS2*, *ATG4D*, *SLC16A1-AS1*	[56,57,59,60]
NRGs	*BCL2*, *JAK3*, *PLA2G4C*, *STAT4*, *CAMK2B*, *PLA2G4C*, *STAT4*, *CASKIN2*, *TLE2*, *USP20*, *SPRN*, *ARSG*, *MIR106B*, *MIR98*, *PLA2G4C*, *STAT4*, *SLC25A6*, *SLC25A4*, *METTL14*, *METTL3*	*CAPN*, *CHMP4C*, *PYGB*, *PLA2G4F*, *CHMP4C*, *TNFSF10*, *ACAT2*, *DHCR7*, *SQLE*, *FDPS*, *MSMO1*, *OSBPL5*, *PLBD1*, *PITPNM3*, *LPCAT2*, *LPCAT4*, *PNPLA3*, *CPNE3*, *SLC44A1*, *SLC2A1*, *PLA2R1*, *ALKBH5*, *HNRNPC*, *WTAP*, *YTHDC2*, *CAPN2*, *CHMP4C*	[61,62,63,64,65,66,67]
PRGs	*MST1*, *ELANE*, *NLRP1*, *AC090114.2*, *AC005332.6*, *PAN3-AS1*, *AC087501.4*, *APIP*, *CHMP6*, *PLCG1*, *SMAD4*, *CDKN2A*	*GSDME*, *GSDMC*, *IL-18*, *NLRP2*, *AC083841.1*, *LINC01133*, *AC015660.1*, *AIM2*, *CASP4*, *CASP6*, *CHMP4C*, *GSDMC*, *GZMB*, *CASP4*, *NLRP1*, *PLCG1*, *IL-18*, *CASP1*, *NLRP2*, *TLR3*, *BAK1*, *TP63*, *CHMP4C*, *PD-L1*	[68,69,70,71,72,73,74,75,76,77,78,79]

**Table 3 biomedicines-11-02792-t003:** List of drugs modulating ferroptosis.

Agent	Mechanisms of Function	Development Stage	Ref.
Cysteinase	XC- system inhibitor	In vivo	[40]
IKE	XC- system inhibitor	In vitro	[40]
Rapamycin	GPX4 depletion	In vivo/in vitro	[97]
RSL3	GPX4 depletion	In vivo/in vitro	[97]
ART	Increasing intracellular ROS and iron accumulation	In vitro	[98]
PL	Increasing intracellular ROS	In vitro	[100]
Ruscogenin	Increasing iron accumulation	In vitro	[99]
ZZW-115	NUPR1 inhibitors, modulation of organelle function (ER and mitochondria), metabolic shifts to glycolysis, suppressing GPX4 and SLC7A11, increasing lipid peroxidation	In vivo/in vitro	[101,102,103]
Zalcitabine	Ferritinophagy	In vivo/in vitro	[104,105]
GEM + chrysin	Inhibition of CBR1, increasing the accumulation of ROS, ferritinophagy	In vivo/in vitro	[106]
GEM + lesinurad	Inhibitor of pan-SLC22A, reduces metastasis	In vivo	[107]
GEM + docosahexaenoic acid	Induced oxidative stress and cell death	In vivo/in vitro	[108]
GEM + EGCG	Inhibition of HSPA5, destabilizing GPX4	In vivo/in vitro	[48]
GEM + SSZ	Inhibition of HSPA5, destabilizing GPX4	In vivo/in vitro	[48]
SSZ + PL + cotylenin A	Accumulation of ROS	In vivo/in vitro	[100]
SSZ + docosahexaenoic acid	Inhibition of SLC7A11, modulating the GSH level, restricting nucleotide synthesis	In vivo/in vitro	[108]
DHA + DDP	Interruption of mitochondrial hemostasis, catastrophic accumulation of free iron, unrestricted lipid peroxidation, degradation of GPX4 and FTH	In vivo/in vitro	[109]
XL888 + anti-PD-1	Inhibition of HSP90, promoting anti-PD-1 inhibitory	In vivo/in vitro	[93]
RSL-3@PVs	Tumor embolisms, inhibiting nutrient delivery, excessive lipid peroxidation, mitochondrial dysfunction	In vivo/in vitro	[110]

**Table 4 biomedicines-11-02792-t004:** List of drugs modulating necroptosis.

Agent	Mechanism of Function	Stage of Treatment	Ref.
SB225002	Inhibition of CXCR2	In vitro	[159]
PDT-MB	Increasing expression of RIPK1, RIPK3, and MLKL	In vitro	[163]
AgNPs	Trigger mixed cell death, disrupt the antioxidant, and induce oxidative and nitro-oxidative mechanisms, increasing RIP-1, RIP-3, and MLKL	In vitro/in vitro	[165,166,167]
IMB5036	Trigger mixed cell death, increasing expression of RIPK1, RIPK3, and MLKL	In vivo/in vitro	[168]
MLN8237 (alisertib)	Inhibition AURKA	In vivo/in vitro	[173]
CCT137690	Inhibition AURKA	In vivo/in vitro	[171]
AdipoRon	Producing superoxide, activating RIPK1	In vivo/in vitro	[174,175]
Vanadium compound	Inhibition of the cell cycle, increasing ROS upregulating RIPK1 and RIPK3	In vitro	[176]
GEM + MLN8237	Increase GEM sensitivity	Clinical (phase I)	[177]
GEM + SK	Regulating the expression of RIP1/RIP3	In vivo/in vitro	[178]
GEM + GAC0003A4	Impair cholesterol and phospholipid metabolism	In vitro	[63]
BV6 + 2′3′-cGAMP	MLKL phosphorylation, stimulation of NF-κB, type I interferons (IFNs), TNFα, and IFN-regulatory factor 1 (IRF1) signaling pathways	In vitro	[179]

**Table 5 biomedicines-11-02792-t005:** List of drugs modulating pyroptosis.

Agent	Mechanisms of Effect	Stage of Treatment	Ref.
VX-765	Inhibition caspase-1	In vivo/in vitro	[68]
NAC	ROS scavenger	In vivo/in vitro	[68]
Dasatinib	Inhibition Src, increasing ceramide levels	In vivo/in vitro	[201]
ceramidase inhibitor B13	Increasing ASAH2 and ceramide levels	In vivo/in vitro	[201]
SEP	Damaging DNA, mitochondrial superoxide anion radical formation induces PANoptosis	In vivo/in vitro	[202,203]
Steroidal saponins PPI/CCRIS/PSV	Caspase-3-mediated cleavage of GSDME	In vivo/in vitro	[204]
LY364947 + ultrasound	TGF-β receptor inhibitor, produced ROS, caused GSDME to be cleaved by caspase-3, broke the dense TME	In vivo	[205]
ALA-lipid/PLGA MBs-mediated SDT	Producing ROS	In vivo/in vitro	[206]
PDT + TBD-3C	Convert cold TME to hot TME	In vivo/in vitro	[207]

## Data Availability

No new data were created or analyzed in this study. Data sharing is not applicable to this article.

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
