# Peer review of "Genome, Metabolism, or Immunity: Which Is the Primary Decider of Pancreatic Cancer Fate through Non-Apoptotic Cell Death?"

_biomedicines, 2023, doi:10.3390/biomedicines11102792_

Round 1

Reviewer 1 Report

In the manuscript the authors describe current knowledge about the importance of non- apoptotic cell death pathways, such as ferroptosis, necroptosis, and pyroptosis and drug resistance in pancreatic ductal adenocarcinoma.

The authors presented this theme based on 204 cited articles, what seem to be adequate to the specific field of the manuscript. The paper is well performed and presented, but I have some comments.

1.     I suggest making a table or a figure summarizing the modulation of non- apoptotic cell death pathways by drugs.

2.     Minor spelling errors need to be corrected:

-       Page 11 line 424 – please change Thereby, By to thereby by

-       Please enter spaces between words and brackets with abbreviations or citations

-       e.g. page 11, line 449, 451,

-       page 13, line 519, 531, 532, 535

Author Response

In the manuscript the authors describe current knowledge about the importance of non- apoptotic cell death pathways, such as ferroptosis, necroptosis, and pyroptosis and drug resistance in pancreatic ductal adenocarcinoma.

The authors presented this theme based on 204 cited articles, what seem to be adequate to the specific field of the manuscript. The paper is well performed and presented, but I have some comments.

We appreciate the positive remarks and encouragement from the reviewer.

  1. I suggest making a table or a figure summarizing the modulation of non- apoptotic cell death pathways by drugs.

We thank the reviewer for this suggestion. We have added tables with a list of drugs modulating non-apoptotic cell death pathways.

  1. Minor spelling errors need to be corrected:

-       Page 11 line 424 – please change Thereby, By to thereby by

-       Please enter spaces between words and brackets with abbreviations or citations

-       e.g. page 11, line 449, 451,

-       page 13, line 519, 531, 532, 535

We apologize for these errors and have corrected these errors. The formatting issues on page 11 and 13 raised by the reviewer is not obvious to us on our review of the manuscript. We suspect this may be due to the format in different word version. Please let us know if this issue persists. 

Reviewer 2 Report

The abstract needs to be changed. It contains  many flaws.

Resistance to apoptosis is not the main resistance mechanism in PDAC. Drug extrusion through P-gp and other membrane proteins of the MDR group is the main mechanism of resistance. Desmoplasia is in second place.

Ferroptosis, pyroptosis do not play any role in PDAC development

Necroptosis is partially apoptotic.  Therefore you cannot say that it is not an apoptotic mechanism.

Your entire abstract is a big collection of errors. You have to write a full new one.

Chemotherapy is not the first line treatment of pancreatic cancer. Surgery is the first line treatment. Only when surgery is not feasible, comes chemotherapy.

It is evident that you do not understand the difference between adjuvant and neoadjuvant chemotherapy.

Neoadjuvant is used before surgery to reduce the size of the tumor or release non-removable structures. Adjuvant comes after surgery.

Gemcitabine is not administered with a folfirinox scheme. These are two different approaches and are not used together. They are alternative schemes not complementary.

Author Response

The abstract needs to be changed. It contains many flaws.

We have revised the abstract.

Resistance to apoptosis is not the main resistance mechanism in PDAC. Drug extrusion through P-gp and other membrane proteins of the MDR group is the main mechanism of resistance. Desmoplasia is in second place.

We agree and have revised the abstract.

Ferroptosis, pyroptosis do not play any role in PDAC development

Thank you for your comment. However, we respectively disagree that ferroptosis and pyroptosis do not play any role in PDAC development. As we discussed in this review, studies have found that ferroptosis and pyroptosis play important roles in PDAC development. Examples include releasing DAMPs induced by ferroptosis and activating immune responses in TME which is known to influence tumorigenesis and PDAC development. Drug-induced ferroptosis is also an emerging strategy to suppress tumor growth in established PDAC [1]. Please refer to reference [2] regarding the role of ferroptosis in PDAC. Additional references [3-7] also attested to the role of ferroptosis and pyroptosis in PDAC.

Necroptosis is partially apoptotic.  Therefore you cannot say that it is not an apoptotic mechanism.

Thank you for your comment. We agree that necroptosis and apoptosis share some common features. However, there are also morphological and biochemical differences between them [8]. Please also refer to Figure 1 in the publication from Karlowitz and colleagues showing the different and shared features between necroptosis and apoptosis [9].

Your entire abstract is a big collection of errors. You have to write a full new one.

We have revised the abstract.

Chemotherapy is not the first line treatment of pancreatic cancer. Surgery is the first line treatment. Only when surgery is not feasible, comes chemotherapy.

We completely agree with the reviewer’s comment, and we apologize if this was not clear in the article. We have revised the wording to say “Most symptomatic PDAC patients are diagnosed at the terminal or metastatic stage when more than 80% of tumors are not resectable, and treatment options are palliative rather than curative [10-14]. As a result, chemotherapy plays a crucial role in treating PDAC,…” Please refer to lines 31-33.

It is evident that you do not understand the difference between adjuvant and neoadjuvant chemotherapy.

Neoadjuvant is used before surgery to reduce the size of the tumor or release non-removable structures. Adjuvant comes after surgery.

The reviewer’s point is well-taken and we apologize for the confusion. We have revised the wording to “As a result, chemotherapy plays a crucial role in the treatment of PDAC, whether administered as monotherapy or as part of adjuvant or neoadjuvant therapy [11, 12].” Please refer to lines 33-35.

Gemcitabine is not administered with a folfirinox scheme. These are two different approaches and are not used together. They are alternative schemes not complementary.

We thank the reviewer to point out our oversight. We have revised the wording to “During advanced stages of PDAC, gemcitabine (GEM) in combination with nanoalbumin-bound paclitaxel (nab-PTX) or FOLFRINOX (a combination of leucovorin, 5-FU, irinotecan, and oxaliplatin) are the cornerstones of chemotherapy based on patients' conditions [8, 10].” Please refer to lines 35-38.

Reviewer 3 Report

Specific comments to the authors

The submitted review " Genome, Metabolism, Or Immunity: Which Is the Primary Decider of Pancreatic Cancer Fate Through Non-apoptotic Cell Death?" rigorously presents and discusses the role of the non-apoptotic cell death processes ferroptosis, necroptosis, and pyroptosis in pancreatic ductal adenocarcinoma (PDAC) based on previously published reviews as well as in vitro, in vivo, and in situ experiments and also in clinical studies.

The topics presented range from the classical aspect of PDAC to important interactions of ferroptosis, necroptosis and pyroptosis with signalling pathways and immunogenetics that may be useful for new treatment options in PDAC. In conclusion, the review provides a very interesting and structured overview of the non-apoptotic cell death processes ferroptosis, necroptosis and pyroptosis in PDAC, which is generally easy to read, follow and understand. The authors should add some aspects before accepting the manuscript for publication, as mentioned below.

 # Table 1: The table is largely simplified and too superficial. Please specify the terms "immune nature" and "lytic". Please replace "character change" with "major morphological change". Please clarify the different pathway of classical apoptosis. Therefore, the authors should intensively revise this table in more detail.

 # Regarding each subchapter "Risk model in PDAC prognosis", the authors should transfer the detected genes/proteins for ferroptosis, necroptosis and pyroptosis into separate tables to highlight these risk factors and identify the "key player" at a glance.

 # Regarding each subchapter "Combination therapy", the authors should transfer the combination therapy to a separate table to identify the "best" combination by additionally showing the experimental effects of these ferroptosis-, necroptosis- and pyroptosis-related drugs in combination therapy.

 # The chapter "Conclusions and perspectives" should give an outlook on how the knowledge could eventually be translated into clinical application. Which ferroptosis-, necroptosis- and pyroptosis-related drugs are suitable for use in humans? Finally, the authors should give a possible answer to the question in the title of their manuscript.

Minor editing of English language required

Author Response

The submitted review " Genome, Metabolism, Or Immunity: Which Is the Primary Decider of Pancreatic Cancer Fate Through Non-apoptotic Cell Death?" rigorously presents and discusses the role of the non-apoptotic cell death processes ferroptosis, necroptosis, and pyroptosis in pancreatic ductal adenocarcinoma (PDAC) based on previously published reviews as well as in vitro, in vivo, and in situ experiments and also in clinical studies.

The topics presented range from the classical aspect of PDAC to important interactions of ferroptosis, necroptosis and pyroptosis with signalling pathways and immunogenetics that may be useful for new treatment options in PDAC. In conclusion, the review provides a very interesting and structured overview of the non-apoptotic cell death processes ferroptosis, necroptosis and pyroptosis in PDAC, which is generally easy to read, follow and understand. The authors should add some aspects before accepting the manuscript for publication, as mentioned below.

We thank the reviewer for the kind words and positive feedback.

 # Table 1: The table is largely simplified and too superficial. Please specify the terms "immune nature" and "lytic". Please replace "character change" with "major morphological change". Please clarify the different pathway of classical apoptosis. Therefore, the authors should intensively revise this table in more detail.

We thank the reviewer for the input. We have revised Table 1 accordingly and clarified different pathways of apoptosis and non-apoptotic cell death.

 # Regarding each subchapter "Risk model in PDAC prognosis", the authors should transfer the detected genes/proteins for ferroptosis, necroptosis and pyroptosis into separate tables to highlight these risk factors and identify the "key player" at a glance.

We thank the reviewer for the suggestion. we have included a new table with lists of important non apoptotic cell death-related genes (Tables 2, page 6).

 # Regarding each subchapter "Combination therapy", the authors should transfer the combination therapy to a separate table to identify the "best" combination by additionally showing the experimental effects of these ferroptosis-, necroptosis- and pyroptosis-related drugs in combination therapy.

Thank you for your comment. There are limited studies focusing on combination therapies and studies used different in vitro or in vivo preclinical animal models. Therefore, it is challenging to identify the “best” combination by these limited studies. Nevertheless, we have included new tables with lists of drugs that modulating each non-apoptotic pathway (Tables 3-5).

 # The chapter "Conclusions and perspectives" should give an outlook on how the knowledge could eventually be translated into clinical application. Which ferroptosis-, necroptosis- and pyroptosis-related drugs are suitable for use in humans? Finally, the authors should give a possible answer to the question in the title of their manuscript.

We thank the reviewer for the suggestion. We have revised the paragraph to incorporate these comments (Page 22-23). As we mentioned above, these studies are in vitro or in vivo animal studies without clinical trial. There is no clear indication for which drugs are suitable for human use and future studies are required to address this question.

Comments on the Quality of English Language

Minor editing of English language required

We have reviewed and revised the English language.

Reviewer 4 Report

The manuscript titled, “Genome, Metabolism, Or Immunity: Which Is the Primary Decider of Pancreatic Cancer Fate Through Non-apoptotic Cell Death?” is interesting and well written. But there are certain aspects missing in this manuscript. In this manuscript, the authors have missed out on elaborating the involvement of autophagy in modulating the metabolism and immunity aspects of PDAC. The authors need to discuss the recently published works involving the role of autophagy in controlling iron metabolism as shown by PMID: 37075122, PMID: 35771492.  In PMID: 37075122, authors have also highlighted the significance of tumor microenvironment, particularly the involvement of cancer associated fibroblasts in modulating iron metabolism in PDAC. Authors need to tabulate the clinical and preclinical findings of non-apoptotic mechanisms involved in PDAC.

Author Response

The manuscript titled, “Genome, Metabolism, Or Immunity: Which Is the Primary Decider of Pancreatic Cancer Fate Through Non-apoptotic Cell Death?” is interesting and well written. But there are certain aspects missing in this manuscript. In this manuscript, the authors have missed out on elaborating the involvement of autophagy in modulating the metabolism and immunity aspects of PDAC. The authors need to discuss the recently published works involving the role of autophagy in controlling iron metabolism as shown by PMID: 37075122, PMID: 35771492.  In PMID: 37075122, authors have also highlighted the significance of tumor microenvironment, particularly the involvement of cancer associated fibroblasts in modulating iron metabolism in PDAC. Authors need to tabulate the clinical and preclinical findings of non-apoptotic mechanisms involved in PDAC.

We thank the reviewer’s valuable feedback. Indeed, autophagy has been recognized as a significant factor in the progression of PDAC. We also noted that autophagy plays a pivotal role in the metabolism and immunity of PDAC. We have included discussions of autophagy in iron metabolism and mitochondria function (lines 224-233). We also included autophagy in other parts of the discussion throughout the review (pages 4,5,7,8). However, due to this limited review, we decided to focus on ferroptosis, necroptosis, and pyroptosis.

Round 2

Reviewer 2 Report

The main cause of the increased incidence of pancreatic cancer is the increase in average population age. Increased fat diet has not been accepted as a cause of pancreatic cancer, and the evidence in this regard is feeble.

When you write

As a result, chemotherapy plays a crucial role in treating PDAC, whether administered as monotherapy or as part of adjuvant or neoadjuvant therapy

Seems that you do not fully understand the difference.

In general, nowadays, PDAC is not treated with monotherapy. Neoadjuvant therapy has two different objectives:

1) limited to the preoperative period in which it represents an intent to make operable tumors that seem borderline for surgery. 

2) in the preoperative cases as a general policy, because it has shown better overall survival post-surgery.

FERROPTOSIS

If you read in detail the reference 34 Tang D, Chen X, Comish PB, Kang R. The dual role of ferroptosis in pancreatic cancer: a narrative review. Journal of Pancreatology. 2021;4(02):76-81.

you will find that the mechanism used by the authors to induce a tumor in mice through ferroptosis consists of experimental manipulation of the antioxidant system ( SLC7A11 or GPX4 depletion). There is no evidence that this can happen in vivo in human PDAC. It is an experiment with transgenic animals. Therefore, it is a poor support of your idea that ferroptosis is a cause of pancreatic cancer in humans. You can mention it in your paper as a hypotheses, but not as evidence. 

Your sentence 

Most symptomatic PDAC patients are diagnosed at the terminal or metastatic stage when more than 80% of tumors are not resectable, and treatment options are palliative rather than curative

Is blatantly incorrect

When patients are terminal or have metastasis 100% of tumors are non-resectable.

The correct sentence should be:

In symptomatic patients 80% are beyond surgical possibilities whether because of distant metastasis or involvement of vital structures.

The following sentence requires a reference

DNAJB11 is a co-chaperone for HSPA5 with a dual effect on PDAC progression. DNAJB11 could regulate the epidermal growth factor (EGFR) expression and initiates the subsequent mitogen-activated protein kinase (MAPK) signaling pathway, ultimately promoting cancer. 

The abbreviation of epidermal growth factor should be EGF.

If you are referring to its receptor you should correct it.

In table 3 I suppose SSZ stands for sulfasalazine. Please clear the issue.

You say

Although inducing ferroptosis has improved cancer treatment, many open questions remain.

As far as I know, ferroptosis induction is still an experimental issue and I do not know any protocol using ferroptotic drugs in clinical oncology. Therefore, please give an example and references of cases in which ferroptosis has improved cancer treatment.

No comments

Author Response

The main cause of the increased incidence of pancreatic cancer is the increase in average population age. Increased fat diet has not been accepted as a cause of pancreatic cancer, and the evidence in this regard is feeble.

We have revised the sentence to read “The number of cases of pancreatic ductal adenocarcinoma (PDAC), the majority prevalent form of pancreatic cancer, is anticipated to increase, mainly due to the average population age increasing.”

When you write

As a result, chemotherapy plays a crucial role in treating PDAC, whether administered as monotherapy or as part of adjuvant or neoadjuvant therapy

Seems that you do not fully understand the difference.

In general, nowadays, PDAC is not treated with monotherapy. Neoadjuvant therapy has two different objectives:

1) limited to the preoperative period in which it represents an intent to make operable tumors that seem borderline for surgery. 

2) in the preoperative cases as a general policy, because it has shown better overall survival post-surgery.

We thank the reviewer for the comment. We have revised to sentence to read “As a result, chemotherapy plays a crucial role in treating PDAC.”

FERROPTOSIS

If you read in detail the reference 34 Tang D, Chen X, Comish PB, Kang R. The dual role of ferroptosis in pancreatic cancer: a narrative review. Journal of Pancreatology. 2021;4(02):76-81.

you will find that the mechanism used by the authors to induce a tumor in mice through ferroptosis consists of experimental manipulation of the antioxidant system ( SLC7A11 or GPX4 depletion). There is no evidence that this can happen in vivo in human PDAC. It is an experiment with transgenic animals. Therefore, it is a poor support of your idea that ferroptosis is a cause of pancreatic cancer in humans. You can mention it in your paper as a hypotheses, but not as evidence. 

We have revised the first paragraph in Ferroptosis to state what the cited studies have found and added animal model to the description.

Your sentence 

Most symptomatic PDAC patients are diagnosed at the terminal or metastatic stage when more than 80% of tumors are not resectable, and treatment options are palliative rather than curative

Is blatantly incorrect

When patients are terminal or have metastasis 100% of tumors are non-resectable.

The correct sentence should be:

In symptomatic patients 80% are beyond surgical possibilities whether because of distant metastasis or involvement of vital structures.

We have revised the sentence according to the reviewer’s suggestion.

The following sentence requires a reference

DNAJB11 is a co-chaperone for HSPA5 with a dual effect on PDAC progression. DNAJB11 could regulate the epidermal growth factor (EGFR) expression and initiates the subsequent mitogen-activated protein kinase (MAPK) signaling pathway, ultimately promoting cancer. 

We have added a reference.

The abbreviation of epidermal growth factor should be EGF.

If you are referring to its receptor you should correct it.

We have changed “epidermal growth factor” to “epidermal growth factor receptor”.

In table 3 I suppose SSZ stands for sulfasalazine. Please clear the issue.

We have added the full name of SSZ (sulfasalazine).

You say

Although inducing ferroptosis has improved cancer treatment, many open questions remain.

As far as I know, ferroptosis induction is still an experimental issue and I do not know any protocol using ferroptotic drugs in clinical oncology. Therefore, please give an example and references of cases in which ferroptosis has improved cancer treatment.

We agree with the reviewer and have added “ in animal models” after “ cancer treatment”.

Reviewer 4 Report

Authors have clarified all my concerns

Author Response

Thank you very much for reviewing our article. 

Round 3

Reviewer 2 Report

No further comments

No further comments